# Influence of Aerodynamic Preloads and Clearance on the Dynamic Performance and Stability Characteristic of the Bump-Type Foil Air Bearing

Fabian Walter *[ID] and Michael Sinapius [ID]

Institute of Mechanics and Adaptronics, Technische Universität Braunschweig, Langer Kamp 6, 38106 Braunschweig, Germany; m.sinapius@tu-braunschweig.de
* Correspondence: f.walter@tu-braunschweig.de

**Abstract:** The dry lubricated bump-type foil air bearing enables a carrying load capacity due to a pressure build up in a convergent air film. Since the air bearing provides low power dissipation above the lift-off speed and the flexible foil provides an adaptivity against high temperatures, manufacturing errors or rotor growth, the bump-type foil air bearing is in particular suitable for high speed rotating machineries. The corresponding dynamic behavior depends on the operational parameters, the behavior of the flexible foil structure, and in particular on the circumferential clearance. In order to avoid or suppress the critical subsynchronous motion at high rotational speeds, many researchers recommend adding an aerodynamic preload to the bore shape, representing a transition from a circular to a lobed bearing bore shape. In addition to positive effects on the stability, preliminary studies demonstrated degrading effects on the stiffness and damping due to increasing preload values. This observation leads to the assumption, that the preload value meets an optimum with respect to stability, load-capacity, and lift-off speed. With the aim of deriving an appropriate lobe configuration for the design of the bump-type foil air bearing, this work performs comprehensive numerical investigations on the dynamic performance and the stability characteristic as a function of preload and minimum clearance. To this end, this work uses steady-state and transient stability analysis methods to recommend optimal aeroydnamic preload values with respect to the corresponding minimum clearance.

**Keywords:** gas bearing; bump-type foil air bearing; lobed bearing; preload; clearance; stability; transient analysis; perturbation analysis; subsynchronous motion

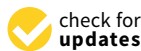



## 1. Introduction

Among sustainable power units, the hydrogen fuel cell has gained importance, containing a high speed rotating air compressor. For the reasons of longevity, the system requires contamination-free bearings, excluding commonly used roller bearings or oil slider bearings. Thus, the dry lubricated bump-type foil air bearing represents a promising machine element because of its low production expenses and low power dissipation. The drawbacks include a crucial range of frictional wear below lift-off speed, poor damping values of the lubricant air film, and resulting self-excited whirl motion at high rotational speeds. Many researchers attribute a stabilizing effect to the preload factor of lobed foil air bearings in order to minmize the corresponding subsynchronous whirl motion. Depending on the number of segments, the aerodynamic preload transforms the bearing bore from a circular to a lobed shape, resulting in higher pressure peaks, which yields a fixation of the journal within the bearing.

*On the Stability and Dynamic Performance of Lobed Bump-Type Foil Air Bearings*

Heshmat et al. [1] established a three-pad bump-type foil air bearing with a decreasing stiffness of the corrugated foil towards the free end and offers further enhancements in [2]

by introducing a multistage corrugated bump-foil, resulting in an aerodynamic preload due to higher foil displacements at the free end and thus in a high whirl stability. Kim [3] readdresses the idea of the aerodynamic preload by introducing a lobed three-pad bump-type foil air bearing with regard to a higher whirl-stability and concluded that preloaded three pad bearings have a decisively reduced load capacity, feature high cross-coupling stiffnesses, but show moreover higher onset speeds of instability compared to circular bearings. In order to increase stability, the authors of [4–6] provide experimental and linear computational studies on the mechanical preloads in bump-type foil air bearings, using shims under the bump-foil. Based on a steady-state, linear computation method, the shimmed bearing shows higher lubricant film stiffness and damping and minor cross-coupled stiffness values, resulting in a more stable performance [4]. The shimmed bearing furthermore tends to a lower minimum film thickness, a higher journal eccentricity, and higher foil deflections, when looking at the stationary bearing quantities [4]. Corresponding *speed-up* and *speed-down* experiments in [5] overall show a lower and delayed subsynchronous response by using shimms. Furthermore, Sim et al. [7–9] experimentally investigate the influence of the aerodynamic preloads on the stability characteristic of the bump-type foil air bearings by testing different lobed configurations. While Ref. [8] considers a constant nominal clearance, and thus a constant compressibility $\Lambda$, the studies in [7,9] investigate the dynamic behavior with increasing preload and constant minimum clearance values. To predict the stability characteristic, the authors apply a linear eigenvalue analysis, based on the dynamic coefficients of the perturbation approach, which underestimates the experimentally specified onset speed of subsynchronous motion (OSS) [8]. In contrast to the shimmed bearings, increasing preloads with a constant minimum clearance lead to a lower film stiffness and damping, which is due to the lower compressibility. Increasing minimum clearances furthermore cause subsynchronous motion and lower rotational speeds of instability. Due to a higher system damping, the increased aerodynamic preload in strongly lobed bearing configurations delays the OSS or completely suppresses self-excited motion. Recent conceptual studies furthermore use the influence of the aerodynamic preload to adapt the dynamic performance to the existing operational parameters. In this context, amongst other authors, Sadri et al. [10,11] and Feng et al. [12] apply actuation systems on the bearing housing, gradually transfering the bearing bore shape from a circular shape to a lobed shape during the rotor speed-up. Consequently, while maintaining a higher minimum film thickness, a lower lift-off speed and a higher load capacity, the adaptive foil air bearing adjusts to a circular bore shape at low rotational speeds. In order to stabilize the journal motion at increasing rotational speeds and to suppress subsynchronous motion, the adaptive bearing adds an aerodynamic preload gradually to its bore shape [10]. Hence, the benefits of different bearing configurations are combined in one adaptive concept.

Altogether, preliminary studies show an enhanced stability characteristic and suppressed subsynchronous whirl motion due to increased preload and decreased minimum clearance values. However, from a bearing design point of view, an optimization, recommending decisive parameter sets of minimum clearance and preload, is still required. Since Sim et al. [9] demonstrated a reduction in the damping values of the bearing as preload increases, it is necessary to check whether the stability enhances constantly with increasing preloads or whether an optimal preload value can be found, that meets an optimum with respect to stability. Furthermore, increasing the aerodynamic preload by using shims, adjusting the bore shape or considering a constant nominal clearance and thus a constant compressibility $\Lambda$, yields a reduction of the minimum clearance. Consequently, preliminary studies often consider preload and minimum clearance in a coupled relationship. Under this condition, it can not be fully distinguished whether the stability is enhanced due to increased preloads or decreased minimum clearances. Both parameters have an evident influence on the stability of the system, and their stabilizing effects must be considered separately. To this end, the following study in particular highlights the stability influence of the preload factor $r_p$ and the minimum clearance $c_m$ on the bump-type foil air bearing, using a steady-state (linear) and a transient (nonlinear) prediction method. The main objec-

tive of the present study is to identify an ideal aerodynamic preload to the corresponding minimum clearance value with respect to stability, load capacity, and lift-off speed. In addition to comprehensive parameter studies on the influence of the preload factor on the dynamic performance of the bump-type foil air bearing, a parameter optimization with respect to the stability, based on an eigenvalue analysis, is given. The linear and the transient procedure are furthermore compared to show the benefits of the linear stability analysis.

## 2. Bump-Type Foil Air Bearing Model

Figure 1 depicts the bump-type foil air bearing, consisting of three flexible, corrugated bump-strip layers, each covered by a top-foil, the bearing shaft, and the housing. The dynamic motion of the journal is determined in Cartesian coordinates $y$ and $x$ by the journal eccentricity $e = \sqrt{y^2 + x^2}$ and attitude angle $\gamma = \arctan \frac{y}{x}$. Furthermore, the bearing journal is loaded by the static force $f_0$. The non-dimensional circumferential clearance $C_0(\theta_{ij})$, describing the distance between the journal and the foil at each circumferential node $\theta_{ij}$, represents the lubricant film at a centric journal position according to Equation (1) (with $C_0 = c_0/c_{nom}$ and $c_{nom} = r_p + c_m$):

$$C_0(\theta) = 1 - \frac{r_p}{c_{nom}} \cos(n_{seg}(\theta - \theta_p)/2) \tag{1}$$

A resulting laminar air flow in axial and circumferential direction, caused by a rotation and an eccentric displacement of the shaft, leads to a dynamic pressure build-up at a converging air film. A corresponding equilibrium of flowrates within the foil air bearing is governed by the isothermal *Reynolds equation* according to Equation (2), which is formed by the *Navier–Stokes-* and the *continuity equation* [13,14] (with $P = p/p_a$, $H = h/c_{nom}$, $Z = z/R$). According to Equation (2), the Reynold's equation is coupled with the nondimensional film function $H(z,\theta)$, according to Equation (3), depending on the circumferential clearance $C_0(\theta)$ (see Equation (1)), the eccentric journal position, and the foil deflection $U(z,\theta)$. Consequently, the dynamic behavior of the foil is considered within the film function $H(z,\theta)$ (see Equation (3)), resulting in a fully coupled system of the foil behavior, the air film, and the rotor motion. The existing prediction methods in this work use the clearance model based on the minimum clearance $c_m$ and the preload factor $r_p$ by Lee et al. [15], which has been successfully compared to a bore shape measurement. Here, the preload factor represents the offset between the radial center of the lobed foil segments and the geometrical bearing bore center, further denoted by the *aerodynamic preload* in a lobed foil air bearing. In addition to that, the preload factor in the present model can be seen as the distance between the circumferential point $\theta_p$ on the pad, with respect to the circumferential segment center, and the minimum clearance circle (see Figure 1). Hence, this study adds the aerodynamic preload to the bore shape, based on three lobed foil pads with a uniform stiffness distribution according to the Case 3 bearing in [3]. The following parametric study compares different lobed configurations with different minimum clearance and preload values, to indicate optimal preload values to the corresponding minimum clearance and consequently to indicate optimal lobe configurations. To represent the lobe configuration of the bearing, the lobe ratio $r_p/c_m$ is introduced, since it fully separates the quantities of preload and minimum clearance.

$$\underbrace{\frac{\partial}{\partial \theta}\left(PH^3\frac{\partial P}{\partial \theta}\right) + \frac{\partial}{\partial Z}\left(PH^3\frac{\partial P}{\partial Z}\right)}_{\text{Poiseuille part}} = \underbrace{\Lambda\frac{\partial}{\partial \theta}(PH)}_{\text{Couette part}} + \underbrace{2\Lambda v\frac{\partial}{\partial \tau}(PH)}_{\text{Instationary part}}, \Lambda = \frac{6\eta\Omega}{p_a}\left(\frac{R}{c_{nom}}\right)^2 \tag{2}$$

$$H(Z,\theta) = C_0(\theta) + \epsilon_x \cos(\theta) + \epsilon_y \sin(\theta) + U(Z,\theta), \text{ with } \epsilon = e/c_{nom} \tag{3}$$

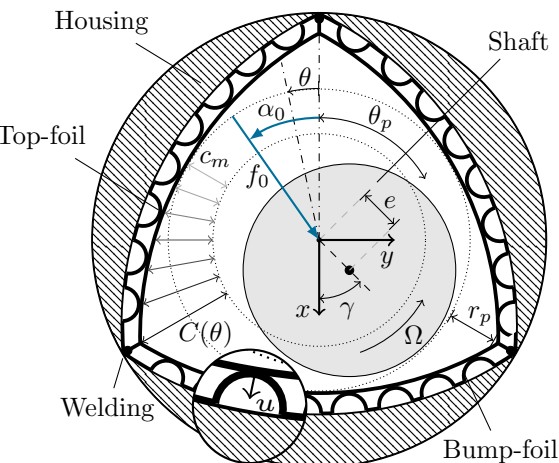

**Figure 1.** Bump-type foil air bearing model according to Sadri et al. [16]

In order to model the corrugated foil, a nonlinear prediction approach from Le Lez et al. [17] is used, which considers the bump-foil as a lattice structure with elementary stiffnesses. Coulomb friction between bump-foil and housing and bump- and top-foil is considered, while the top-foil stiffness is disregarded. Taking into account the global stiffness matrix, a decoupled, equivalent structural stiffness $k_{st}(\theta_{ij})$ and damping $c_{st}(\theta_{ij}) = \frac{\beta}{v}k_{st}(\theta_{ij})$, based on the structural loss factor $\beta$ and the frequency ratio $v$, can be established. Based on the structural stiffness $k_{st}$ and damping $c_{st}$, the reaction force $f_{st}$ of the bump-foil can be obtained according to Equation (4):

$$f_{st}(\theta) = k_{st}(\theta)u(\theta) + c_{st}(\theta)\dot{u}(\theta) \tag{4}$$

The validation of the foil model in Figure 2, based on the displacement results of a cantilever beam in a uniform and a increasing load case (LC) by Le Lez et al. [17], shows a good agreement. The present NDOF bump-foil model, established by Le Lez et al. [17], is further extended by Arghir et al. [18,19], which is not the scope of the present study.

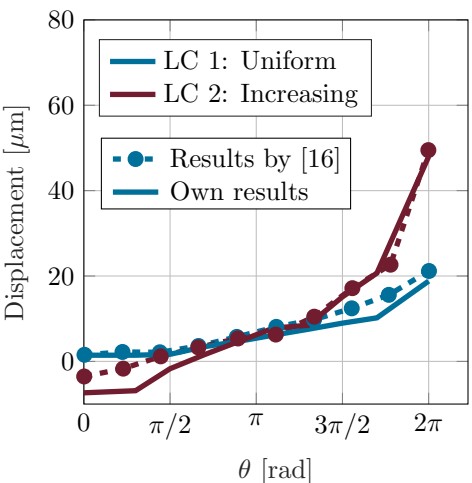

**Figure 2.** Validation of the foil model under consideration of different load cases (LC) at a cantilever beam, set up by Le Lez et al. [17].

## 3. Methods of Solution

### 3.1. Steady-State Analysis

A numerical prediction scheme, using the *Finite-Difference Method* (FDM) according to Heshmat et al. [20], can be carried out to solve the 0th order isothermal Reynold's equation

and to calculate the nondimensional steady-state pressure $P_0$ within the bearing. Since the nondimensional film function $H(\theta, Z)$ and the Reynold's equation are in a coupled relationship, the numerical solution of the pressure has to be iterated until the minimum film thickness, including the foil deflection, converges. Starting from the computed steady-state pressure distribution $P_0$, the corresponding nondimensional lubricant reaction force $F_b$ can be written as

$$\begin{bmatrix} F_{bx} \\ F_{by} \end{bmatrix} = \frac{D}{4L} \int_0^{2\pi} \int_{-L/D}^{L/D} (P_0 - 1) \begin{bmatrix} \cos(\theta) \\ \sin(\theta) \end{bmatrix} dZ d\theta \tag{5}$$

Under the condition of an equilibrium between the static bearing load $F_0 = f_0 / p_a D L$ and the lubricant reaction force $F_b = f_b / p_a D L$, the nondimensional eccentricity $\epsilon = e/c_{nom}$ and the attitude angle $\gamma$ are updated, using a *Newton-Raphson* iteration approach following Equation (6), until the equilibrium exists. Each iteration step of eccentricity and attitude angle performs an updated pressure and film thickness iteration loop.

The results of the steady-state quantities of the validation case bearing (**VC**, see Table 1) are validated in Figure 3, using the results of Sim et al. [9]. Further numerical considerations on the FDM, boundary conditions and mathematical procedures are given in [16,20–22]:

$$\epsilon_{n+1} = \epsilon_n - (F_{bn} - F_0) / \frac{F_{bn} - F_{bn-1}}{\epsilon_n - \epsilon_{n-1}} \text{ and } \gamma_{n+1} = \gamma_n - (\alpha_{bn} - \alpha_0) / \frac{\alpha_{bn} - \alpha_{bn-1}}{\gamma_n - \gamma_{n-1}} \tag{6}$$

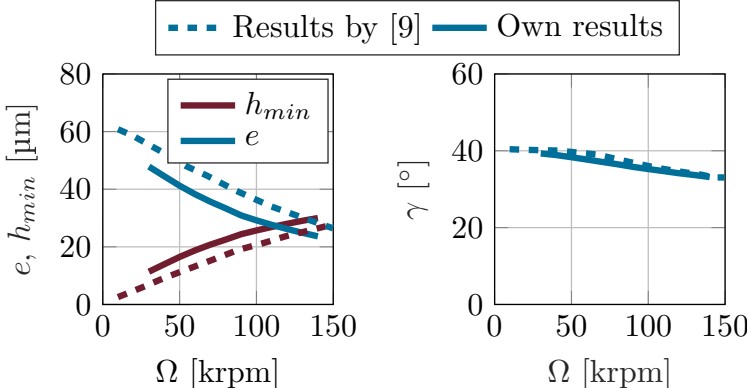

**Figure 3.** Validation of the steady-state eccentricity, minimum film thickness and attitude angle, based on results of Sim et al. [9], with respect to the **VC** bearing with $r_p = 100$ µm.

**Table 1.** Bump-type foil air bearing properties of the test case **TC 1** and **TC 2** and the validation case bearings **VC** according to [9].

| Parameter | TC 1 | TC 2 | VC |
|---|---|---|---|
| **Bore shape** | | | |
| $c_m$ [μm] | 50 | 30 | 50 |
| $r_p$ [μm] | 0 to 600 | 0 to 360 | 0, 100 |
| $D$ [mm] | 38.5 | | 25.6 |
| $L$ [mm] | 40 | | 25.3 |
| **Bump foil** | | | |
| $n_{seg}$ [-] | 3 | | 3 |
| $\theta_p$ [°] | 60 | | 60 |
| $n_b$ [-] | 9 | | 6 |
| $R_b$ [mm] | 2 | | 2 |
| $l_0$ [mm] | 1.81 | | 1.2 |
| $s$ [mm] | 4.57 | | 2.7 |
| $t_b$ [mm] | 0.127 | | 0.12 |
| $E$ [GPa] | 213 | | 214 |
| $\nu$ [-] | 0.29 | | 0.29 |
| $\mu$ [-] | 0.5 | | 0.1 |
| $\beta$ [-] | 0.3 | | 0.4 |
| **Operational parameters** | | | |
| $f_0$ [N] | 5 | | 1.9 |
| $\alpha_0$ [°] | 0 | | 0 |
| $v$ (steady-state) [-] | 1 | | 1 |
| $\rho$ (transient) [gmm] | 1 | | - |

Although the aerodynamic foil air bearing describes a highly nonlinear system, several authors established linearization methods in order to approximate the dynamic behavior at a defined operational point, especially depending on the rotational speed $\Omega$, the static load $f_0$, and the excitation frequency ratio $v$, using a set of dynamic coefficients of stiffness and damping ($k_{xx}, k_{xy}, k_{yx}, k_{yy}, c_{xx}, c_{xy}, c_{yx}, c_{yy}$). Considering small translocations $\epsilon$ and translocation velocities $\dot{\epsilon}$ within a small perturbed journal motion the authors of [20,23] perform partial derivations of the lubricant reaction forces with the translocation and the translocation velocity, based on the steady-state 0th order Reynolds equation, to obtain the colinear and cross-coupled dynamic coefficients. A numerically more accurate perturbation approach for general gas bearings, established by Lund [24,25] and applied on the bump-type foil air bearing by Peng et al. [14], is obtained by linearizing the instationary 1st order Reynolds equation, using a Taylor expansion. The assumed harmonic oscillation of the perturbed journal around the static equilibrium can be written in a complex formulation, according to [3,26], as $\Delta\hat{\epsilon}_{x,y} = Re\{|A|_{x,y}e^{i\tau}\}$, $\Delta\dot{\hat{\epsilon}}_{z,y} = i\Delta\hat{\epsilon}_{x,y}$ and leads to the perturbation terms of pressure $\hat{P}$, film thickness $\hat{H}$, and foil deflection $\hat{U}$ symbolized by $\hat{Q}$, as given in Equation (7). As a result of linearizing the transient 1st order Reynold's equation, using the Taylor expansion, the Reynold's equation in a perturbed condition can be set up. This yields to a system of equations, consisting of a steady-state term with respect to the steady-state pressure $P_0$ (see Equation (8)) and two first order perturbed terms in $y$- and $x$- directions. The perturbed terms are summarized in Equation (9) with respect to the perturbed pressure quantities $\hat{P}_x$ and $\hat{P}_y$. Numerically solving the perturbed terms and computing $\hat{P}_x$ and $\hat{P}_y$, using the FDM, allows to calculate the dynamic coefficients according to Equation (10). In terms of a validation, Figure 4 compares the calculated dynamic coefficients of the validation case (VC) with the results of Sim et al. [9]. Deviations in the dynamic coefficients, as well as in the steady-state quantities in Figure 3, to the validation data are due to the different foil model. The authors in [9] used Iordanoff's [27] formula, constituting a simplified approach to calculate the compliance of single free-end or fixed bumps:

$$\hat{Q} = \hat{Q}_0 + \Delta\epsilon_x\hat{Q}_x + \Delta\epsilon_y\hat{Q}_y \tag{7}$$

$$\frac{\partial}{\partial \theta}\left(P_0 H_0^3 \frac{\partial P_0}{\partial \theta}\right) + \frac{\partial}{\partial Z}\left(P_0 H_0^3 \frac{\partial P_0}{\partial Z}\right) = \Lambda \frac{\partial}{\partial \theta}(P_0 H_0) \tag{8}$$

$$\frac{\partial}{\partial \theta}\left(\hat{P}_{x,y} H_0^3 \frac{\partial P_0}{\partial \theta} + 3\hat{H}_{x,y} P_0 H_0^2 \frac{\partial P_0}{\partial \theta} + P_0 H_0^3 \frac{\partial \hat{P}_{x,y}}{\partial \theta}\right) +$$
$$\frac{\partial}{\partial Z}\left(\hat{P}_{x,y} H_0^3 \frac{\partial P_0}{\partial Z} + 3\hat{H}_{x,y} P_0 H_0^2 \frac{\partial P_0}{\partial Z} + P_0 H_0^3 \frac{\partial \hat{P}_{x,y}}{\partial Z}\right) = \Lambda\left(\frac{\partial}{\partial \theta} + 2vi\right)(\hat{P}_{x,y} H_0 + P_0 \hat{H}_{x,y}) \tag{9}$$

$$\frac{c_{nom}}{p_a L D}\left(\begin{bmatrix} k_{xx} & k_{yx} \\ k_{xy} & k_{yy} \end{bmatrix} + i\omega\Omega\begin{bmatrix} c_{xx} & c_{yx} \\ c_{xy} & c_{yy} \end{bmatrix}\right) = \frac{D}{4L}\int_0^{2\pi}\int_{-L/D}^{L/D}\begin{bmatrix} \hat{P}_x\cos(\theta) & \hat{P}_x\sin(\theta) \\ \hat{P}_y\cos(\theta) & \hat{P}_y\sin(\theta) \end{bmatrix}dZd\theta \tag{10}$$

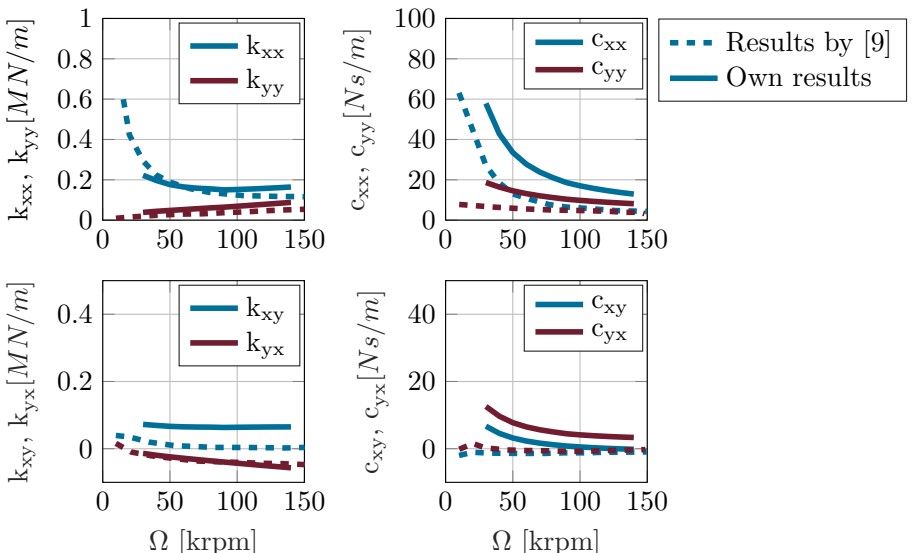

**Figure 4.** Validation of the synchronous dynamic coefficients, based on results of Sim et al. [9], with respect to the **VC** bearing with $r_p$= 100 µm.

With regards to a linear stability analysis, several authors like [3,6,25,28,29] refer to the nondimensional stability criterion, established by Pan [30], evaluating the modal impedance of the gas bearing. This study evaluates the stability through the determination of the system damping ratio $D_{sys}$, using an eigenvalue analysis of the linear rotordynamic system according to Equation (11). In this way, the 4th order characteristic polynomial in Equation (12), expressed in a scalar product and based on the solution approach $[x,y]^T = [\hat{x}, \hat{y}]^T e^{\lambda t}$, requires a numerical solution. The system damping is represented by the real part and the eigenfrequency by the imaginary part of the computed eigenvalue (Equation (13)). Since the eigenvalue analysis is based on the linear dynamic coefficients, resulting from a linearization, the analysis is further denoted as *linear stability analysis*. Instability occurs at operational points that have zero or negative damping. Thus, excitation in the eigenfrequency consequently leads to a critical increase in the amplitude. Due to the lack of damping, entered energy can not be dissipated. Consequently, positive damping values of $c_{xx}$ and $c_{yy}$ dissipate energy, while Lund [25] emphasized that positive $k_{xy}$ values and negative $k_{yx}$ values provide negative system damping and supply energy to the journal motion. Referring to this, only the forward whirl leads to unstable motion, since this case provides negative damping [25]. Consequently, it can be summarized that cross-coupling stiffness and damping values represent a source of instability [29].

$$m_r \begin{bmatrix} 1 & 0 \\ 0 & 1 \end{bmatrix} \begin{bmatrix} \ddot{x} \\ \ddot{y} \end{bmatrix} + \begin{bmatrix} c_{xx} & c_{xy} \\ c_{yx} & c_{yy} \end{bmatrix} \begin{bmatrix} \dot{x} \\ \dot{y} \end{bmatrix} + \begin{bmatrix} k_{xx} & k_{xy} \\ k_{yx} & k_{yy} \end{bmatrix} \begin{bmatrix} x \\ y \end{bmatrix} = f_0 \begin{bmatrix} \sin(\alpha_0) \\ \cos(\alpha_0) \end{bmatrix} + \rho \Omega^2 \begin{bmatrix} \sin(\Omega t) \\ \cos(\Omega t) \end{bmatrix} \quad (11)$$

$$\begin{bmatrix} m_r^2 \\ m_r(c_{yy} + c_{xx}) \\ m_r(k_{yy} + k_{xx}) + c_{xx}c_{yy} - c_{xy}c_{yx} \\ c_{xx}k_{yy} + k_{xx}c_{yy} - c_{xy}k_{yx} - k_{xy}c_{yx} \\ k_{yy}k_{xx} - k_{xy}k_{yx} \end{bmatrix} \cdot \begin{bmatrix} \lambda^4 \\ \lambda^3 \\ \lambda^2 \\ \lambda^1 \\ \lambda^0 \end{bmatrix} = \mathbf{0} \quad (12)$$

$$f_{eig} = \frac{Im\{\lambda\}}{2\pi}, D_{sys} = -\frac{Re\{\lambda\}}{\omega_0} \quad (13)$$

### 3.2. Transient Analysis

The nonlinear, transient approach, on the other hand, calculates journal displacements based on lubricant film-induced reaction forces for individual timesteps by numerically solving the equation of motion of the bearing-rotor system, which enables orbit simulations and the mapping of nonlinear phenomena such as self-excited, subsynchronous journal motions that lead to instabilities at higher rotational speeds. Due to this fact, the according analysis is further denoted as *nonlinear stability analysis*.

An investigation with respect to stability of lobed bump-type foil air bearings by Kim [3] involves nonlinear orbit simulations and shows decisively higher onset speed of instability than the steady-state analysis, based on the impedance formulation. In order to investigate the dynamic behavior of the bump-type foil air bearing, Le Lez et al. [31] extend the foil model, presented in Section 2, to a dynamic, transient model. Transient orbit simulations show that the dry friction of the structural deflection as a dissipative element enhances the stability characteristic of the bearing, compared to rigid bearings.

The authors of [10,32] perform a transient solution of a symmetrical rotor-bearing system in time, which agrees well with the steady-state, linear method [32]. Based on an equal rotor-bearing system, Bonello et al. [33] and Baum et al. [34] present the transient *Galerkin-Reduction* (GR) prediction approach with reduced numbers of state variables, which provides much improved computational efficiency compared to the *Finite-Difference-Method* (FDM). In addition to orbit simulations, Baum et al. [35] furthermore apply a bifurcation and *Poincaree* analysis in order to indicate subsynchronous or unstable motion. Using a Bubnov–Galerkin weak formulation of the instationary Reynold's equation to numerically calculate the pressure distribution, Larsen et al. [36] set up a coupled nonsymmetrical bearing-rotor system under consideration of gyroscopic effects.

Furthermore, Guo et al. [37] developed a nonlinear, transient bearing-rotor system, considering the instationary lubricant reaction forces of the bump-type foil air bearing, unbalance excitation, and gyroscopic effects of the rotor. A comparison of the simulative and experimental coast-down shows that the rotary whip and whirl occur in the natural eigenfrequencies of the bearing. The study also provides essential parametric studies regarding the bump stiffness, nominal clearance, static load, and structural loss factor of a cylindrical bearing.

In terms of the transient, nonlinear approach, this study applies the FDM according to Sadri et al. [10], considering the system by the six state variables of displacement, translocation velocity, foil deflection, and pressure, which form the state vector $\psi = [x, \dot{x}, y, \dot{y}, p, u]$ according to Equation (14). This state system consequently represents a symmetrical, rigid rotor, mounted on two identical bump-type foil air bearings, under consideration of a synchronous unbalance excitation $\rho \Omega^2$ and the static load $f_0$. Starting from a defined eccentric point of the journal, the transient approach first calculates the corresponding steady-state quantities of foil deflection and pressure distribution until the minimum film thickness $H_{min}$ converges, as shown in Section 3.1. The resulting steady-state quantities of eccentricity, translocation velocity, foil deflection, and pressure are assigned

as the initial values of the following transient analysis. Subsequently, the nondimensional time derivatives of the state system are calculated in each time step and are simultaneously solved using the explicit *Adams–Bashfort* method, a numerical solver for stiff differential equation systems [38]. Accordingly, each time-step calculates the instationary lubricant reaction forces, based on the integration of the instationary pressure distribution of the time-step according to Equation (5). The lubricant forces further allow the calculation of the derivative of the second and fourth state variable, the translocation acceleration. Solving the first order Reynolds equation, based on the pressure distribution of the current time-step, leads to the time derivative of the pressure $\dot{p}(\theta)$. Particularly, to calculate the foil deflection of the subsequent time-step, the corresponding time derivative $\dot{u}(\theta)$ needs to be calculated in the current time-step, which is derived from the formulation of the bump-foil reaction force in Equation (4).

$$
\begin{bmatrix} \dot{\psi}_1 \\ \dot{\psi}_2 \\ \dot{\psi}_3 \\ \dot{\psi}_4 \\ \dot{\psi}_5 \\ \dot{\psi}_6 \end{bmatrix} = \begin{bmatrix} \psi_2 \\ \frac{1}{m_r}\left(f_{bx}(t) + f_0 \sin(\alpha_0) + \rho\Omega^2 \cos(\Omega t)\right) \\ \psi_4 \\ \frac{1}{m_r}\left(f_{by}(t) + f_0 \cos(\alpha_0) + \rho\Omega^2 \sin(\Omega t)\right) \\ \dot{p}(\theta) \\ \dot{u}(\theta) = \frac{1}{c_{st}(\theta)}\left(f_{st}(\theta) - k_{st}(\theta)u(\theta)\right) \end{bmatrix}
\tag{14}
$$

The eccentricity of the linear prediction approach constitutes a steady-state solution, considering synchronous and asynchronous excitation. In contrast, the nonlinear, transient method maps a non-stationary journal motion, iteratively converging to a steady-state orbit. Consequently, the mean value of the transient orbit over all timesteps $n$ yields the nonlinear eccentricity of the corresponding stationary steady-state. In order to identify subsynchronous journal motion, a *Fast-Fourier Transformation* (FFT) will be carried out, using a Hanning window.

## 4. Parametric Study

In the following parametric study, the influence of the preload factor on the dynamic performance and the stability characteristic of the **TC 1** and **TC 2** bearings, given in Table 1, are investigated. Within the parameter study, the preload factor is increased with a constant minimum clearance $c_m = 50$ μm (**TC 1**) and $c_m = 30$ μm (**TC 2**). In order to fully understand the influence of the aerodynamic preload on the dynamic performance and stability, this study covers a wide range of lobed bearing configurations from $r_p/c_m = 0$ to $r_p/c_m = 12$. Since the stiffness distribution of the foil is assumed to be uniform, the foil properties are kept constant according to the foil parameters in Table 1. Assuming a rotor weight of 1000 g, the static load on a single bearing satisfies 5 N. The rotor is in particular exposed to a harmonic, synchronous unbalance excitation, which is introduced with a constant value of $\rho = 1$ gmm in the transient analysis. Regarding the steady-state analysis, the synchronous dynamic coefficients with a frequency ratio of $v = \omega/\Omega = 1$ consequently apply for the unbalance excitation case.

### 4.1. Dynamic Performance

First, indications of the bearing performance are provided by the dynamic coefficients with respect to the aerodynamic preload, shown in Figure 5. Increasing preload values result in a decreasing stiffness and damping. Cylindrical or slightly lobed bearing configurations with low preload also contain strongly positive $k_{xy}$ and negative $k_{yx}$ values that add energy to the motion [25]. As preload values increase, the cross-coupling coefficients of stiffness and damping converge to zero, resulting in minor cross-coupling effects of the bearing. Adding an aerodynamic preload to the bore shape of the foil air bearing apparently leads to an overall softening of the lubricant film. Since the colinear damping values decrease, but the critical cross-coupling coefficient values converge to zero, an effect on the stability characteristic can not be deduced and requires further investigations.

Furthermore, this observation leads to the assumption that an optimum of the preload factor with respect to whirl stability exists. In addition to a stiffening of the lubricant film at higher rotational speeds, a decisive decrease in the lubricant damping can be observed, which reduces the ability to suppress critical self-excited amplitudes.

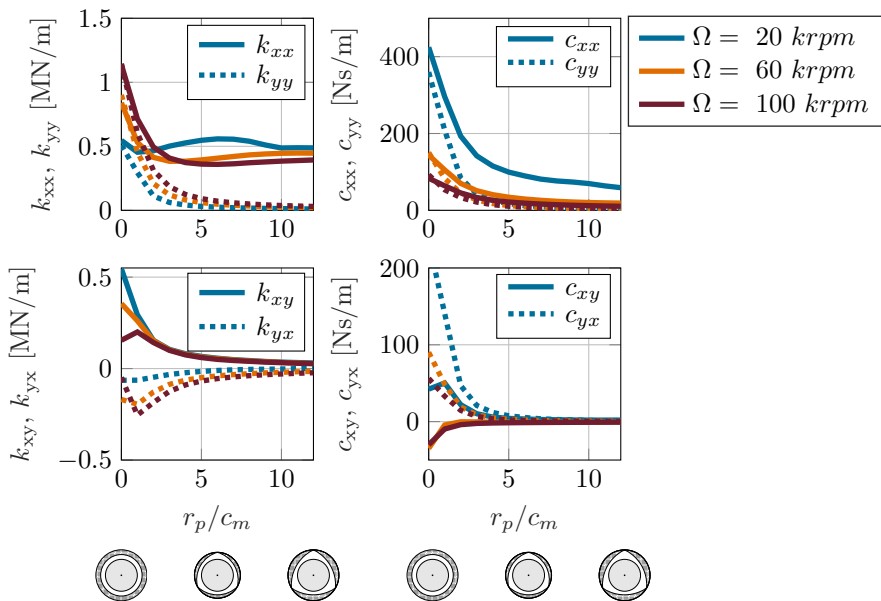

**Figure 5.** Synchronous dynamic coefficients of **TC 1** as a function of $r_p/c_m$ at different rotational speeds.

The softening of the lubricant film, as a result of an increasing preload or minimum clearance, leads to a lower load capacity, as can be seen in Figure 6. In this context, the analysis of the maximum load capacity assumes a maximum eccentricity of $e = 0.9 \cdot c_m$ and an attitude angle $\gamma = 0°$. With increasing preload and minimum clearance, the bearing journal tends to operate at higher eccentricities, resulting in a lower minimum film thickness and consequently in a lower lift-off speed (Figure 6). The stiffening of the lubricant film, as a result of increasing rotational speeds, leads to a centering of the journal position and thus to a lower stationary eccentricity. As a result of the lubricant film properties, it can be further summarized that cylindrical bump-type foil air bearings provide a significantly better lift-off characteristic than lobed bearings, which leads to lower wear at low rotational speeds.

On the other hand, the bore shape has a significant influence on the attitude angle $\gamma$ of the journal. As the preload factor increases, the attitude angle decreases to a lower limit, considering a constant rotational speed. However, the attitude angle of the journal increases with an increasing minimum clearance. This characteristic is due to the shape of the pressure distribution in the circumferential direction. The higher the preload factor $r_p$, the more pronounced is the pressure peak in load direction, which reduces the attitude angle of the journal in the circumferential direction (Figure 7).

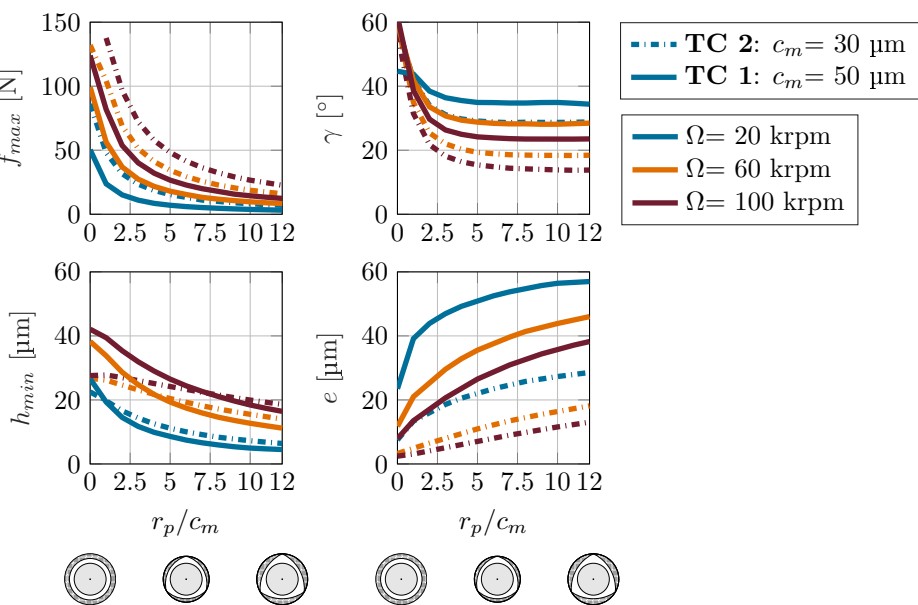

**Figure 6.** Maximum load capacity, eccentricity, minimum film thickness, and attitude angle as a function of $r_p/c_m$ of **TC 1** and **TC 2** at different rotational speeds.

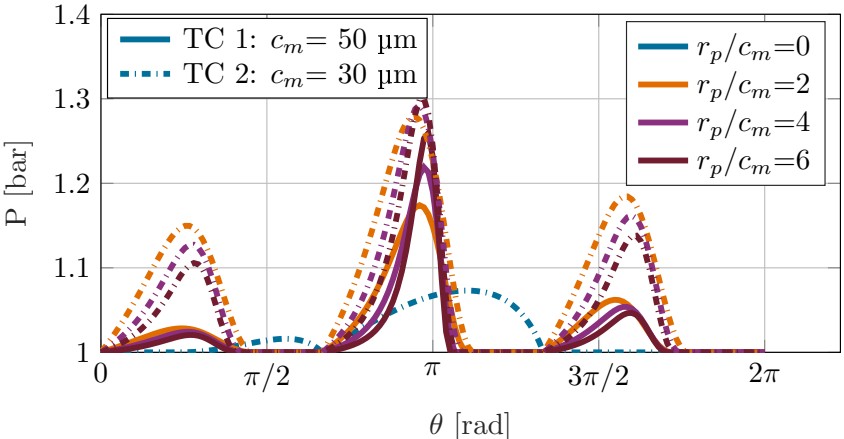

**Figure 7.** Pressure distribution of **TC 1** and **TC 2** in circumferential direction $\theta$ at $\Omega$ = 60,000 rpm within the axial bearing center.

The pressure distribution in Figure 7 furthermore highlights the effect of the aerodynamic preload, generating a fixation of the journal between the pressure peaks of each segment.

Figure 8 shows the maximum foil deflection within the bearing in the circumferential direction at increasing rotational speeds and a constant static load of $f_0 = 5$ N. The maximum foil deflection of the **TC 1** bearing shows a converging behavior, while the foil deflection of the **TC 2** bearing decreases to a minimum and further increases at higher rotational speeds. Furthermore, the influence of the compliant bump foil directly affects the dynamic behavior of the bearing and and can be observed in the trend of the dynamic coefficients in Figure 5. In contrast to the stiffness $k_{yy}$ transverse to the load direction, the stiffness $k_{xx}$ in load direction at lower rotational speeds exceeds the stiffness at higher rotational speed as preload increases. As shown before, especially at low rotational speeds, the journal operates at high eccentricities, close to the minimum clearance circle and consequently at its loading limit. Due to the emerging lubricant reaction forces, the foil deflection increases decisively, raising the film thickness and stiffness. This characteristic dynamic behavior of the bump-foil highlights the promoting influence of the flexible structure, maintaining

a carrying load capacity in limit load cases. More heavily lobed configurations exhibit a higher maximum foil deflection, forming a higher pressure peak in load direction and consequently a stronger focussed load distribution. Due to higher foil deflections, the dynamic behavior of the foil has a more significant contribution to the dynamic behavior of the bearing in stronger lobed configurations.

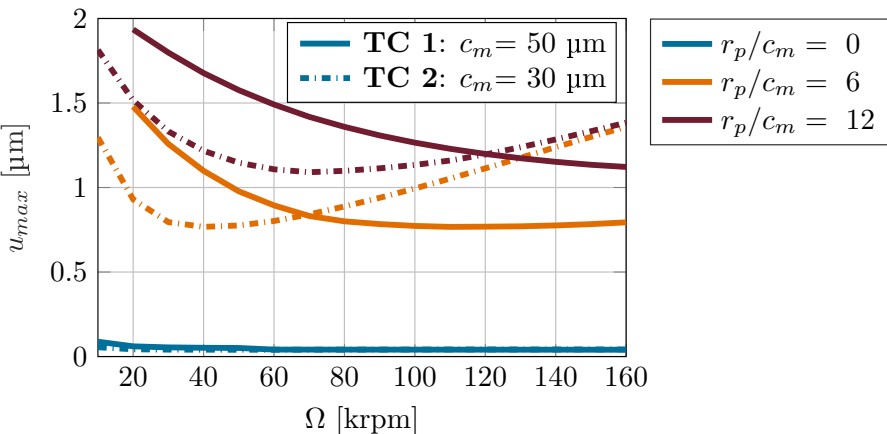

**Figure 8.** Maximum foil deflection of **TC 1** and **TC 2** in different lobe configurations as a function of rotational speed.

### 4.2. Stability Analysis

This section investigates the influence of the preload factor $r_p$ on the stability, in order to provide a resulting parameter optimization with respect to the preload and minimum clearance and intends to show the opportunities and limitations of the linear stability prediction method. Figure 9 depicts the stationary eccentric point and attitude angle of the journal during a run-up, comparing the steady-state and transient approach of **TC 1**. Both methods predict a decrease in journal eccentricity with increasing rotational speeds, caused by a stiffening of the lubricant film, indicating a good agreement. Since the transient analysis additionally considers the dynamic pressure, depending on the translocation velocity of the journal, the journal tends to lower eccentricities in the transient approach, compared to the steady-state method. Furthermore, the nonlinear method predicts an onset speed of instability of the $r_p/c_m = 2$ configuration at $\Omega \approx 105{,}000$ rpm. The journal appears to perform an unstable motion, leading to critical minimum film thicknesses and causing the collapse of the lubricant film [38]. In terms of predicting the stationary eccentric position of the journal within the bearing, the linear and nonlinear methods agree well.

Consequently, Figure 10 depicts the damping ratios and the eigenfrequencies of the 1st and 2nd eigenmodes. Due to the decreasing lubricant film damping (Figure 5), the damping ratio overall decreases in the 1st and 2nd eigenmode as rotational speed increases. Negative damping values occur in the lighter lobed configurations $r_p/c_m = 2$ and $r_p/c_m = 3$ at $\Omega = 40{,}000$ rpm and $\Omega = 90{,}000$ rpm. Furthermore, the damping ratios of the lobed bearings with $r_p/c_m = 4$ and $r_p/c_m = 6$ are positive at every operational point of rotational speed in the 1st eigenmode. However, the damping ratio curve of $r_p/c_m = 4$ is slightly above the higher lobed configuration with $r_p/c_m = 6$ (see Figure 10).

Increasing the preload factor consequently shifts the damping ratio of the 1st eigenmode into positive ranges. While the system damping ratios of the more lightly lobed cases decrease in the negative range, the ratios of the more heavily lobed cases constantly remain at low positive values.

When looking at the 2nd eigenmode, a decreasing damping ratio becomes apparent according to increasing preload.

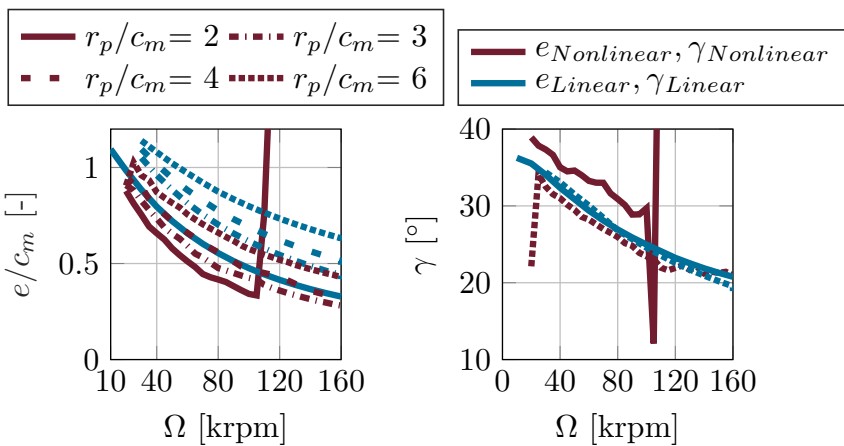

**Figure 9.** Steady-state linear and nonlinear eccentricity $e$ and attitude angle $\gamma$ of the shaft of **TC 1** as a function of rotational speed.

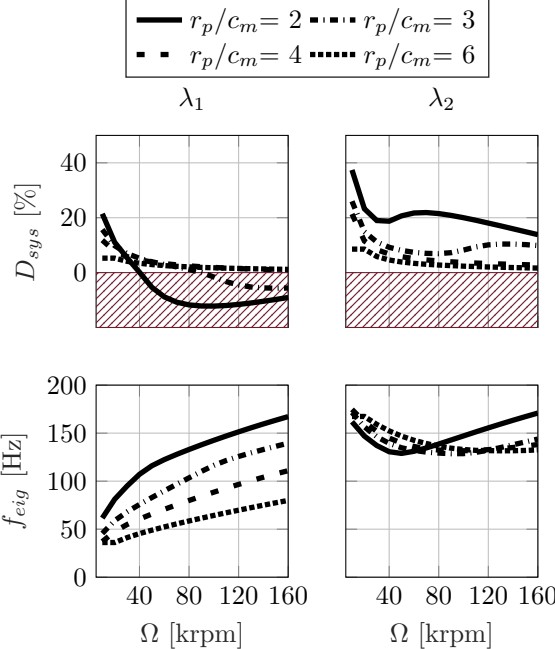

**Figure 10.** System damping ratio and eigenfrequency of the 1st and 2nd eigenmode of **TC 1** of different lobed configurations as a function of rotational speed.

In order to allow considerations of the journal dynamics in the time and frequency domain, the transient prediction method needs to be taken into account. Thus, Figure 11 shows waterfall plots of a run-up simulation of different, preloaded configurations ($r_p/c_m = \{2, 3, 4, 6\}$). Furthermore, this figure highlights the journal orbit in the time domain at $\Omega$ = 20,000 rpm, the median and the maximum rotational speed, providing a qualitative measure.

Looking at the waterfall plots and thus the journal displacement in the frequency domain, the synchronous and the subsynchronous, self-excited amplitudes are to be emphasized. In addition to the synchronous, unbalance excited amplitudes, the journal performs high subsynchronous motion in a horizontal direction (Figure 11, 1st column) and essentially smaller subsynchronous motion in a vertical direction (Figure 11, 2nd column), which is due to the higher stiffness in a vertical load direction (Figure 5). During the run-up, the subsynchronous amplitude increases, which is consistent with the trend of the linear eigenvalue analysis in Figure 10, as the system damping decreases. Consequently, the lubricant film provides less damping at higher rotational speeds to suppress self-excited

vibrations. Depending on the preload configuration, two major subsynchrononous frequencies can already be seen, which are filtered out from the waterfall plots and further investigated in Figure 12. A comparison of the displayed subsynchronous frequencies with the linearly calculated eigenfrequencies initially leads to a good agreement. The estimated eigenfrequencies in the 1st eigenmode of the linear method slightly remain over the 1st subsynchronous frequencies, especially when looking at the stronger lobed configurations. Overall, however, the first subsynchronous frequency coincides with the eigenfrequency of the 1st eigenmode and the second frequency with that of the 2nd eigenmode. Self-excitation obviously occurs close to the eigenfrequencies of the bearing.

Considering the amplitudes according to its lobed shape, the first subsynchronous frequency shows an increasing amplitude in the horizontal direction and a decreasing amplitude in vertical direction, as preload increases.

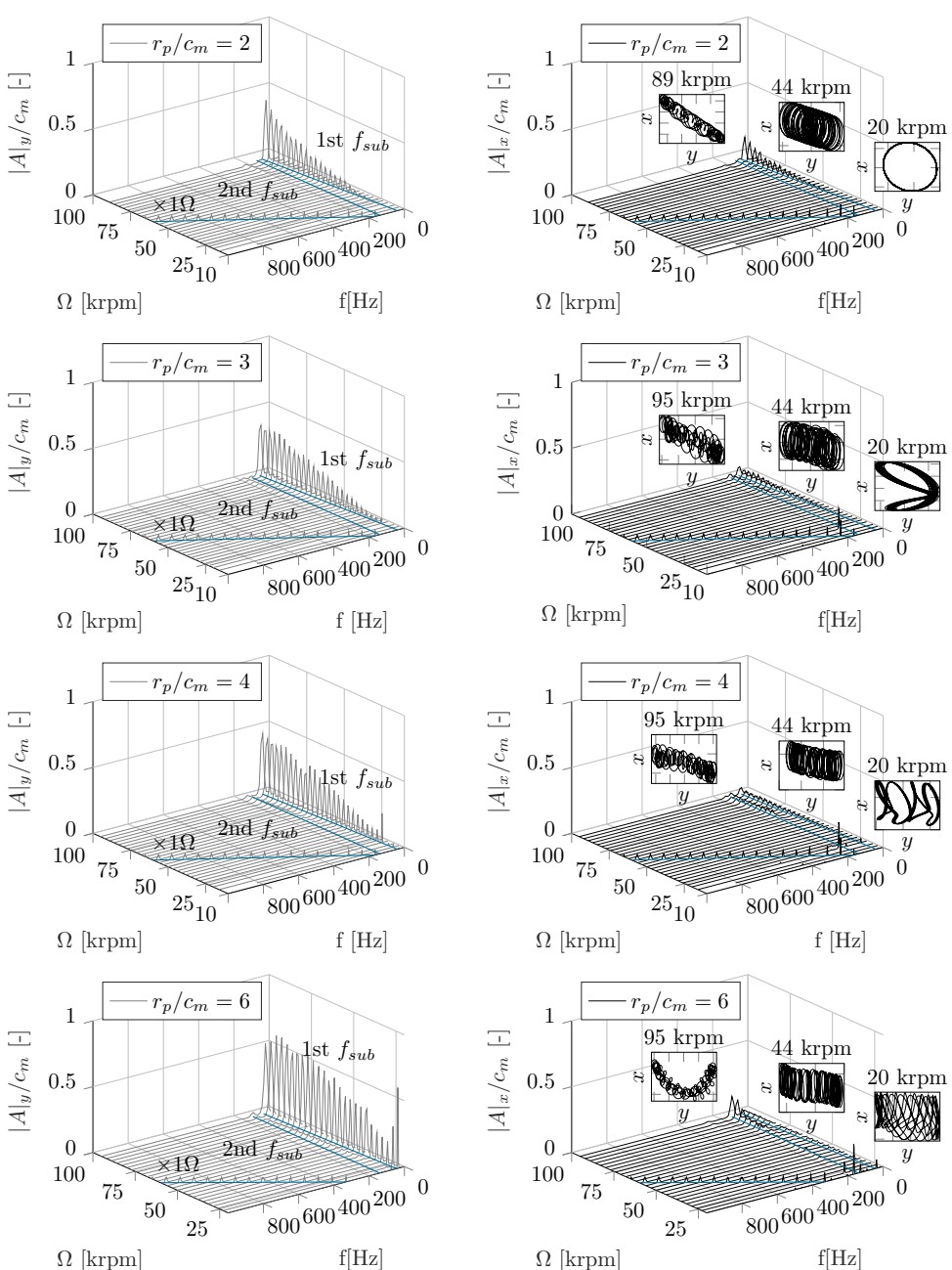

**Figure 11.** Numerically calculated run-up waterfall plots of **TC 1** with different lobe configurations.

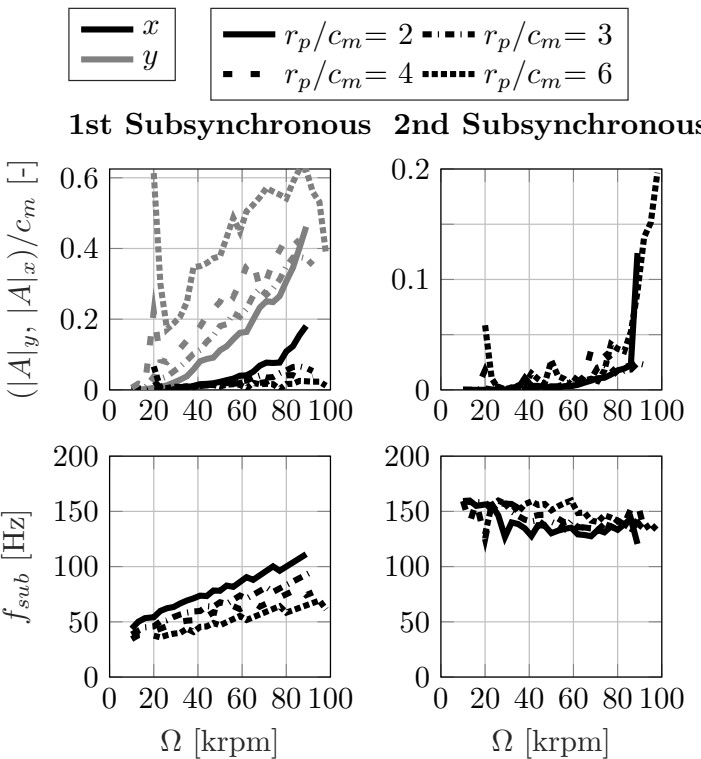

**Figure 12.** Amplitude and frequency of the 1st and 2nd subsynchronous response during rotor run-up of **TC 1** with different lobed configurations.

On the other hand, the amplitude of the 2nd subsynchronous frequency shows critical amplitude raises in the lightest lobed ($r_p/c_m = 2$) and in the most lobed ($r_p/c_m = 6$) configuration. The linear and the nonlinear method agree especially in the frequency range of the 2nd frequency, since both methods predict a decrease in stability. Additionally, the vertical amplitudes of the 1st subsynchronous frequencies agree well with the predicted results of the linear approach, since the vertical amplitude decreases with increasing preload. Furthermore, the onset speed of subsynchronous motion of the 1st frequency in the vertical direction of the ($r_p/c_m = 2$)- and ($r_p/c_m = 3$)- configurations are located closely at the predicted stability limits of the linear analysis.

However, the horizontal response of the 1st subsynchronous frequency decisively exceeds the vertical response and shows a contradiction with the linear results, since the subsynchronous amplitude increases with increasing preload. Moreover, the subsynchronous motion in a horizontal direction, which mainly corresponds to the 1st eigenmode, operates outside the valid range of perturbed motion and therefore can not be represented by the linear analysis.

Regarding the stability, it should be noted that the vertical motion in load direction is much more critical than horizontal motion. Figure 9 shows the static equilibrium of the configuration $r_p/c_m = 6$ at $e \approx 40\ \mu m$, $\gamma \approx 23°$ and $\Omega = 100,000$ rpm, which represents a high eccentric point in a vertical direction. Thus, the vertical displacement contains fewer means until the journal exceeds the minimum clearance and tends to touch the housing.

Consequently, the identification of an optimum lobe configuration requires an appropriate balance in the vertical and horizontal amplitude of the 1st and 2nd subsynchronous frequency. According to Figure 12, the ($r_p/c_m = 3$)- and ($r_p/c_m = 4$)- configurations turn out to be the most balanced ones in terms of the occurrence of the subsynchronous frequencies. Moreover, stronger lobed configurations are limited by poor load capacities and lift-off speeds, which are essential design parameters.

### 4.3. Parameter Optimization

In order to determine an optimum lobe configuration, Figure 13 shows the linear eigenvalue analysis at $\Omega = 100{,}000$ rpm with respect to the lobe ratio $r_p/c_m$, considering different lobe configurations. The optimum configuration is assumed at the highest system damping ratio in the 1st and in the 2nd eigenmode. However, at the optimum, the damping ratio must be positive in both modes to avoid critical subsynchronous amplitudes. As shown in Figure 12, both eigenmodes contribute to the self-excitation of subsynchronous motion. Thus, simply considering the maximum in one single eigenmode is not sufficient. According to the linear eigenvalue analysis, based on the synchronous coefficients of stiffness and damping, Figure 13 shows one characteristic optimum at $r_p/c_m = 4$, with a minimum clearance ranging from 20 µm to 50 µm. This observation in the linear prediction approach is consistent with the dynamic response of the nonlinear method in frequency domain, since this configuration exhibits a balanced dynamic response and maintains stability up to high rotational speeds. Beyond the optimal lobe configuration, a slight decrease of the system damping is observed with increasing preload values.

As the minimum clearance increases, the system damping increases and so does the stability of the bearing. In addition to that, Figure 14 shows the subsynchronous response of the test case bearings **TC 1** and **TC 2** with a preload factor of $r_p = 300$ µm, confirming a decisive reduction of the subsynchronous amplitude by decreasing the minimum clearance. In conclusion, to increase the stability characteristic of the bearing at the optimum lobe configuration, the minimum clearance must be minimized, which decisively increases the system damping. Further increasing the preload does not further increase the stability of the bearing and additionally degrades the load capacity and lift-off speed, as the stationary and transient prediction approaches have shown, which needs to be further experimentally investigated.

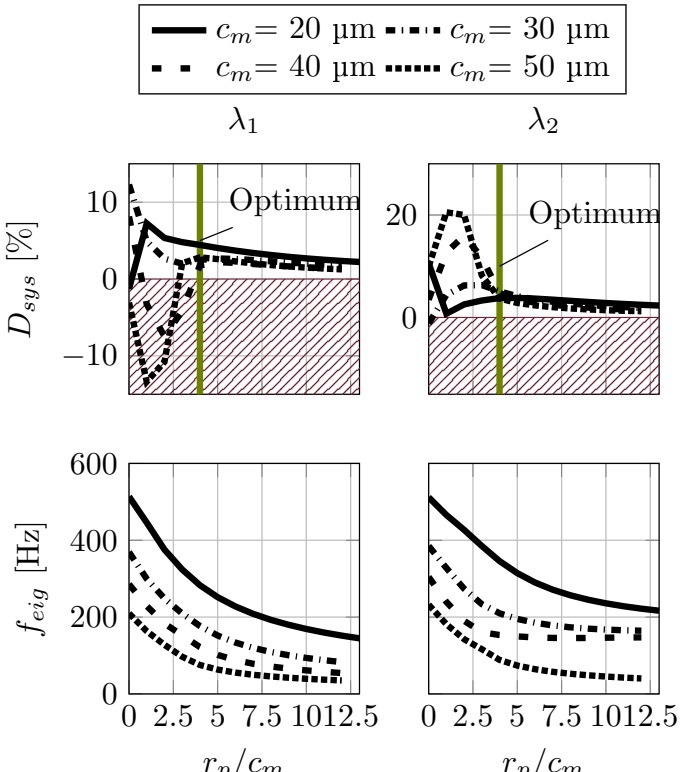

**Figure 13.** System damping ratio and eigenfrequency of the 1st and 2nd eigenmode of **TC 1** as a function of $r_p/c_m$ at $\Omega = 100{,}000$ rpm.

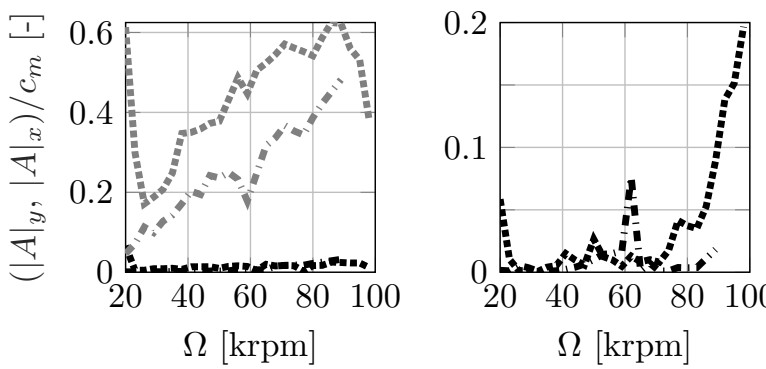

**Figure 14.** Subsynchronous amplitudes of **TC 1** and **TC 2** with $r_p$ = 300 µm.

## 5. Conclusions

The present study investigates the influence of the aerodynamic preload and the minimum clearance in a context with the dynamic performance and the stability characteristic of the bump-type foil air bearing. Accordingly, an increasing preload factor degrades the maximum load capacity, increases the lift-off speed, and results in a lower lubricant film stiffness and damping, but yields fewer cross-coupling effects.

In order to design bump-type foil air bearings with respect to the stability and to reduce subsynchronous whirl motion, this work presents a linear eigenvalue analysis and transient method in the frequency domain. The linear stability analysis indicates an increase of system damping in the 1st eigenmode and a decrease in the 2nd eigenmode as preload increases. A comparison of both methods shows an occurrence of self-excited, subsynchronous motion close to the eigenfrequencies, corresponding to the 1st and 2nd eigenmode of the bearing.

With the scope of designing a stable bearing, while maintaining an appropriate lift-off speed and load capacity, a medium lobed configuration with $r_p/c_m = 4$ is recommended, since this configuration shows the most balanced dynamic response. For this purpose, critical subsynchronous motion in vertical and horizontal directions is predicted in the transient analysis in the frequency domain, related to lighter and much more lobed bearing configurations. Increasing the aerodynamic preload beyond the optimum leads to higher subsynchronous motion in the horizontal and vertical direction of the bearing, which can further be suppressed by reducing the minimum clearance.

The presented findings, based on the numerical studies, need to be experimentally investigated.

**Author Contributions:** Methodology, investigation, computation, and validation, F.W.; software, F.W.; writing, F.W., M.S.; project management, scientific consulting and paper review, M.S. All authors have read and agreed to the published version of the manuscript.

**Funding:** This research was funded by FVV, project number 1267. We also acknowledge support by the German Research Foundation and the Open Access Publication Funds of Technische Universität Braunschweig.

**Institutional Review Board Statement:** Not applicable.

**Informed Consent Statement:** Not applicable.

**Data Availability Statement:** Data is contained within the article.

**Acknowledgments:** The research project (1267) was performed by the Institute of Mechanics and Adaptronics (IMA) at Technical University of Braunschweig under the direction of Ing Michael Sinapius. It was financed by the FVV (Research Association for Combustion Engines eV) and the Schaeffler AG and conducted by an expert group under the direction of Joachim Schmied (Delta JS). The authors gratefully acknowledge the support received from the FVV, the Schaeffler AG and from all those involved in the project. Furthermore, we acknowledge support by the German Research Foundation and the Open Access Publication Funds of Technische Universität Braunschweig.

**Conflicts of Interest:** The authors declare no conflict of interest.

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
