# Peer review of "Influence of Aerodynamic Preloads and Clearance on the Dynamic Performance and Stability Characteristic of the Bump-Type Foil Air Bearing"

_machines, doi:10.3390/machines9080178_

Round 1
Reviewer 1 Report
The article about aerodynamic preloads and clearance on the dynamic performance and stability characteristics of the bump-type foil air bearing may be interesting for researchers and practitioners. I have got several remarks:
- What do authors understand by aerodynamic preload? Is it adding shrimps between the bearing sleeve and bump foil? The article is about its influence on foil bearings, but shrimps influence the foil part of foil bearing and gas part of foil bearing. How was implemented foil part of gas bearings?
- Were the stiffness and damping coefficients of foils changed during the calculations? What were their values?
- In the article is variable “the static load F0(a0)” What is its value and why it is a function of alpha? It is a function of rotational speed?
- The stiffness and damping coefficients showed in Figure 6 are calculated for the gas part of foil bearing, or they are the sum of stiffness and damping coefficients for a gas and a foil part of the foil bearing.
- The abbreviation used by authors “et. al.” should be written without the dot, in this way “et al.”.
- In the article, there is the sentence: “… that positive kxy values and negative kyx values provide negative damping and supply energy to the journal motion.” The stiffness and damping may be connected, but usually one of them doesn’t imply the other one. Maybe the authors write something more on this subject.
- It would be good to add some literature about analysing foil bearings, for example, https://doi.org/10.3390/app11020878.
- What does this sentence mean: “Adding an aerodynamic preload to the bore shape of the foil air bearing apparently leads to an overall softening of the lubricant film”?
- In figure 8 is visible that the foil deflection decreases to a minimum and increases further at higher rotational speeds. What can imply this behaviour?
- Instead of the reference in the text “Pan and Coda [30] should be “Pan [30].
- Figure 8 should be updated in the legend to include all curves on the figure.
Reviewer 2 Report
The paper presents the stability characteristics of preloaded lobe gas bearings. Linear stability assessment is presented applying eigenvalue formulation of a simple system with linearized bearing coefficients of stiffness and damping.
The paper presents some interesting results which may be proved useful for the future design of such preloaded gas bearings. The article is well written and the quality of presentation is high. The formulation is rather simplistic for such a highly nonlinear problem, and the results can be judges as theoretical enough and as preliminary, mainly due the fact that the stability assessment applies local linearization around an equilibrium position. Therefore, a perfectly balanced system is considered; the assumption is rather theoretical as in practice no perfectly balanced rotor applies. An unbalance force may trigger instability at speeds where linear stability analysis proves stable operation. Unstable solution branches may exist close to the stable equilibrium solution branch.
More advanced tools should be applied for the study of the stability in such systems (continuation tools).
The results are presented in dimensioned form at most cases. Usually, in order to help the reader to perceive the scientific outcome, bearing coefficients are presented in dimensionless form as a function of Sommerfeld number or other “bearing number” and further to that, the results of stability threshold are presented as a function of a dimensionless design parameter (design of the bearing or of the rotor); please check relative documentation for parameter Gama (Γ) and dimensionless rotating speed.
At any case the paper is worth publishing and no revisions are required (accept as it is).
[1] Wang, J., Khonsari, M., Application of Hopf Bifurcation Theory to Rotor-Bearing Systems with Consideration of Turbulent Effects, Tribology International 39 (2006) 701-714.
[2] Wang, J., Khonsari, M., Influence of Inlet Oil Temperature on the Instability Threshold of Rotor-Bearing Systems, Journal of Tribology 128 (2006) 319-326.
[3] Miraskari., M., Hemmati, F., Gadala, M., Nonlinear Dynamics of Flexible Rotors Supported on Journal Bearings - Part I: Analytical Bearing Model, Journal of Tribology 140 (2018) 021704.
[4] Chasalevris, A., Stability and Hopf bifurcations in rotor-bearing-foundation systems of turbines and generators, Tribology International 145 (2020) 106154
[5] T. H. Kim, L. San Andres (2008) Heavily loaded gas foil bearings: a model anchored to test data, Journal Engineering for Gas Turbines and Power 130 (1) 012504–012508.
[6] L.S. Andres, Notes 15 Gas Film lubrication
Round 2
Reviewer 1 Report
Most of my remarks were taken into account. Authors reply that "Unfortunately I don’t find any source with this DOI: https://doi.org/10.3390/app11020878". This is the full reference:
Breńkacz, Ł.; Bagiński, P.; Żywica, G. Experimental Research on Foil Vibrations in a Gas Foil Bearing Carried Out Using an Ultra-High-Speed Camera. Appl. Sci. 2021, 11, 878. https://doi.org/10.3390/app11020878.